Genetic diversity and connectivity of the megamouth shark (Megachasma pelagios)

Liu Shang Yin Vanson oceandiver6426@gmail.com 1 2
Joung Shoou Jeng 3 4
Yu Chi-Ju 3
Hsu Hua-Hsun 3 4 5
Tsai Wen-Pei 6
Liu Kwang Ming 4 7
1 Department of Marine Biotechnology and Resources, National Sun Yat-Sen University , Kaohsiung , Taiwan
2 Doctoral Degree Program in Marine Biotechnology, National Sun Yat-Sen University , Taiwan
3 Department of Environmental Biology and Fisheries Science, National Taiwan Ocean University , Taiwan
4 George Chen Shark Research Center, National Taiwan Ocean University , Taiwan
5 Center for Environment and Water, Research Institute, King Fahad University of Petroleum and Minerals , Saudi Arabia
6 Department of Fisheries Production and Management, National Kaohsiung Marine University , Kaohsiung
7 Institute of Marine Affairs and Resource Management, National Taiwan Ocean University , Taiwan
Reimer James
Electronic publication date: 2018 Mar 5
Publication date: 2018
Volume: 6
Electronic Location ID: e4432
Received 2017 Nov 16; Accepted 2018 Feb 9
Copyright: ©2018 Liu et al.
Copyright year: 2018
Copyright holder: Liu et al.
License: This is an open access article distributed under the terms of the Creative Commons Attribution License, which permits unrestricted use, distribution, reproduction and adaptation in any medium and for any purpose provided that it is properly attributed. For attribution, the original author(s), title, publication source (PeerJ) and either DOI or URL of the article must be cited.
License URL: https://creativecommons.org/licenses/by/4.0/

Keywords: Migration, Pelagic shark, Connectivity, Panmictic population, Genetic diversity

Funding: Ministry of Science and Technology (MOST) MOST 104-2313-B-019-002 MOST 106-2611-M-022-001 MOST 106-2611-M-110-009 This study was funded by the Ministry of Science and Technology (MOST), ROC (MOST 104-2313-B-019-002 to Shoou Jeng Joung, MOST 106-2611-M-022-001 to Wen-Pei Tsai and MOST 106-2611-M-110-009 to Shang Yin Vanson Liu). The funders had no role in study design, data collection and analysis, decision to publish, or preparation of the manuscript.

==============================
The megamouth shark (Megachasma pelagios) was described as a new species in 1983. Since then, only ca. 100 individuals have been observed or caught. Its horizontal migration, dispersal, and connectivity patterns are still unknown due to its rarity. Two genetic markers were used in this study to reveal its genetic diversity and connectivity pattern. This approach provides a proxy to indirectly measure gene flow between populations. Tissues from 27 megamouth sharks caught by drift nets off the Hualien coast (eastern Taiwan) were collected from 2013 to 2015. With two additional tissue samples from megamouths caught in Baja California, Mexico, and sequences obtained from GenBank, we were able to perform the first population genetic analyses of the megamouth shark. The mtDNA cox1 gene and a microsatellite (Loc 6) were sequenced and analyzed. Our results showed that there is no genetic structure in the megamouth shark, suggesting a possible panmictic population. Based on occurrence data, we also suggest that the Kuroshio region, including the Philippines, Taiwan, and Japan, may act as a passageway for megamouth sharks to reach their feeding grounds from April to August. Our results provide insights into the dispersal and connectivity of megamouth sharks. Future studies should focus on collecting more samples and conducting satellite tagging to better understand the global migration and connectivity pattern of the megamouth shark.

Introduction

The megamouth shark, Megachasma pelagios, was accidentally captured in 1976 off the coast of Kāne’ohe, Hawai’i, and was examined and described as a new species in 1983 (Taylor, Compagno & Struhsaker, 1983). More than forty years since its discovery, only about 100 specimens have as yet been caught or documented. There are only few official records including a review by Nakaya (2010), which documented 40 records of these sharks being either caught or released from 1976 to 2007. The Ichthyology section of the Florida Museum of Natural History has documented 65 sighting records from 1976 to 2016 (https://www.floridamuseum.ufl.edu/fish/discover/sharks/megamouths/reported-sightings). In addition, with recently added records from Taiwan (34 individuals) and Puerto Rico (one individual) (Hsu et al., 2015; Rodriguez-Ferrer et al., 2017), only 99 individuals have been officially recorded (a global sighting record list based on scientific literature is given in Table S1). To date, relatively few studies have focused on this species compared to other, better known sharks. It is suggested to be a widely distributed species across the world’s oceans, including the Indian, Pacific, and Atlantic. Males become mature at about 4 m in total length and females at about 5 m, and mating occurs all year round based on the record of 40 specimens sampled from 1976 to 2008 (Nakaya, 2010). Their daily movements were recorded by acoustic tags and showed a very clear vertical movement. This vertical movement indicated they swim at depths around 200 m during daytime, move toward the surface at dusk, remain around 20 m during nighttime, and move back down to a deeper layer at dawn (Nelson et al., 1997). This shark feeds exclusively on euphausiids (Taylor, Compagno & Struhsaker, 1983; Yano, Tsukada & Furuta, 1998; Sawamoto & Matsumoto, 2012) and employs engulfment feeding analogous to humpback whales (Nakaya, Matsumoto & Suda, 2008). Their pectoral fins are very flexible and mobile, which enhance dynamic lift control and thus give stability while swimming at slow speed (Tomita et al., 2014). In addition, due to the scarcity and vulnerability of these sharks, satellite tagging has not yet been feasible. Therefore, information about their horizontal movement and migration is still unknown. Among the sharks recorded, only few specimens have been used for genetic studies (i.e., phylogenetic relationships, mitochondrial genome) (Martin & Naylor, 1997; Chang et al., 2013), and most of them were discarded or consumed. Due to its rarity, population studies such as demographics, population structure, and genetic diversity among different geographic regions are difficult to conduct.

The region along the Kuroshio Current path, including the Philippines, Taiwan, and Japan, are the countries where the megamouth shark is frequently found (74 out of 99). The number of documented records from Taiwan (45 out of 99) was the highest in the world.

Taiwan initiated its National Plan of Action concerning sharks in May 2006 (Taiwan Fisheries Agency, 2006) and implemented a ban on shark finning in 2012. Additionally, to monitor the catch of several threatened shark species, the Taiwan Fisheries Agency implemented a mandatory catch and report measurement scheme in 2013 that included the megamouth shark (M. plagios), basking shark (Cetorhinus maximus), and great white shark (Carcharodon carcharias) in addition to the whale shark (Rhincodon typus). When these species are caught, fishers must immediately inform the local Fishery Agency, Taiwan Fisheries Agency, and shark experts (National Taiwan Ocean University) before further processing. Due to this management measure, our team was able to obtain fishery biology information such as total length, body weight, sex, and the relationships between measurements and tissue samples before the sharks were processed and sold (Hsu et al., 2015).

Sharks are facing global decline, and the effects (i.e., lost of genetic diversity) of population decline are of major concern in marine conservation (Pinsky & Palumbi, 2014). Loss of genetic diversity has several potential consequences on reducing evolutionary potential and adaptive ability (i.e., decreasing fitness and resistance) (Frankham, 2005; Allendorf et al., 2008). The objectives of this study were to reveal the genetic diversity and connectivity of the megamouth shark with two tissue samples collected from the Baja California, Mexico; 27 tissue samples from Hualien, eastern Taiwan; and published sequences from GenBank.

Materials and Methods

A total of 27 tissue samples of M. plagios were collected between 2013 and 2015 off Hualien county, eastern Taiwan (Fig. 1). These sharks were caught at night before dawn between April and August and were the bycatch of drift-gillnet fishery. This fishery operated year round, mainly targeting sun fishes during spring and summer and bill fish during fall and winter. Basic information recorded included catch date, sex, body weight, and total length. Additionally, maturity stages were determined by examining the developmental status of sexual organs. Males having fully calcified claspers that twisted easily and fully developed testes and epididymides were determined to be adults. Females with mature ova in their ovaries (both ovaries in the megamouth shark) and having swollen oviducts and uteri were determined to be adults. If only one or portions of these organs were developed, individuals were determined to be sub-adults, and those whose sexual organs were in undeveloped stages were determined to be juveniles. Meanwhile, tissue samples were collected at the harbor before further commercial processing, preserved in 95% alcohol, and stored at 4 °C. In addition to samples collected from Taiwan, we obtained two tissue samples deposited in the Scripps Institute of Oceanography, University of California, San Diego, that were collected from the coastal area of Baja California, Mexico (SIO-07-53, Bahia Tortugas; SIO11-299, Bahia Sebastian Vizcaino). One cox1 sequence downloaded from GenBank was derived from a specimen collected from Mojacasabe Beach, Cabo Rojo, Puerto Rico (17.980570 N, −67.210663 W), and one from Indonesia (Fig. 1).

Figure 1 Sampling sites of the megamouth shark (M. plagios).

An asterisk indicates tissue sample sites and + indicates sequences downloaded from GenBank.

Genomic DNA was extracted from tissue fragments using commercial DNA extraction kits (Geneaid Tissue Genomic DNA mini Kit; Geneaid Biotech, New Taipei City, Taiwan). DNA extracts were diluted in TE buffer and stored at −20 °C until amplification by polymerase chain reaction (PCR).

Amplification of genetic markers

The partial mitochondrial DNA gene cox1 was amplified with the primer pair F1/R1 described by Ward et al. (2005). An additional microsatellite locus (Loc6) that has been successfully cross-amplified in lamniform sharks was also amplified, since it showed a high variation in not only repeat number but also flanking regions (Martin et al., 2002). PCRs were run in 30 μL reactions containing 10–40 ng template DNA, 3 μL 10X buffer, 0.2 mM dNTPs, 1.5 mM MgCl2, 10 mM of each primer, and 0.2 units of Taq polymerase (MDbio, Taipei, Taiwan). The thermocycling profile consisted of initial denaturation at 94 °C for 2 min followed by 35 cycles of denaturation at 94 °C for 30 s, annealing at 55 °C for 30 s, extension at 72 °C for 40 s, and a final extension at 72 °C for 2 min. This program was used to amplify the cox1 gene and Loc6. The nucleotide sequences of PCR products of both loci were determined using an ABI 377 automated sequencer (Carlsbad, CA, USA). Nucleotide sequences were assembled and edited using Geneious 9.1.2 (Biomatters, Auckland, New Zealand).

Genetic analyses

Two cox1 gene sequences of individuals from Indonesia (EU3938905) and Puerto Rico (KY392958.1) were downloaded from GenBank. In addition, a Loc6 sequence derived from a Japanese specimen was downloaded (AF423063) (Fig. 1). Arlequin 3.5 (Excoffier & Lischer, 2010) was used to analyze genetic diversity indexes, including haplotype diversity (h) and nucleotide diversity (π). Sequences were aligned and exported to MEGA 7 (Tamura et al., 2013) to visually inspect all alignments. Phylogenetic analyses were used to reveal potential genetic divergences among specimens from different geographic locations, with maximum likelihood (ML) and Bayesian inference assessments being performed on the CIPRES Science Gateway (Miller et al., 2015) and MrBayes (MB) version 3.2.2 (Ronquist et al., 2012), respectively. The latter implemented two parallel runs of four simultaneous Markov chains for 10 million generations, sampling every 1,000 generations and using default parameters. The first million generations (10%) were discarded as burn-in, based on the stationarity of log-likelihood tree scores. ML analyses were conducted in RAxML version 8.1.24 (Stamatakis, 2014) using the HKY substitution model chosen by MEGA 7. Supporting values on the branch were evaluated by non-parametric bootstrapping with 1,000 replicates performed with RAxML (ML) and by posterior probabilities (MB). Moreover, median-joining haplotype networks were generated based on cox1 and Loc6 sequence datasets by using Popart 1.7 (Leigh & Bryant, 2015).

Results

Catch information

Basic catch information showed that megamouth sharks were mainly caught between April and August, with total weights ranging 210–1,147 kg and total lengths ranging 341–710 cm. The sex ratio (female:male) was 16:11, which was not significantly different from 1:1. Five of the 27 individuals were determined to be adults and the others were sub-adults (Table 1).

Table 1 Biological information and GenBank accession number of megamouth shark samples used in the present study.

Name	Date of collection	Sex	Weight (kg)	TL (cm)	Life stage	cox1	Loc 6	
MP1	2013∕4∕18	F	366	387	Sub-adult	HQ010081	MG461954	
MP2	2013∕4∕30	F	383	373	Sub-adult	MG461955	MG461954	
MP3	2013∕5∕6	F	1,090	476	Adult	–	–	
MP4	2013∕5∕6	M	413	368	Sub-adult	HQ010081	MG461954	
MP5	2013∕5∕7	M	328	385	Sub-adult	–	MG461954	
MP6	2013∕5∕8	F	408	413	Sub-adult	HQ010081	MG461954	
MP7	2013∕5∕18	F	516	524	Sub-adult	MG461955	–	
MP8	2013∕5∕18	F	452	552	Sub-adult	HQ010081	MG461954	
MP9	2013∕5∕19	M	320	395	Sub-adult	HQ010081	MG461954	
MP10	2013∕5∕21	M	320	363	Sub-adult	HQ010081	MG461954	
MP11	2013∕5∕30	F	516	426	Sub-adult	MG461956	MG461954	
MP12	2013∕6∕13	M	348	380	Sub-adult	HQ010081	MG461954	
MP13	2013∕7∕10	F	549	463	Sub-adult	HQ010081	MG461954	
MP14	2013∕7∕10	F	348	398	Sub-adult	HQ010081	MG461954	
MP15	2013∕7∕10	M	653	484	Adult	HQ010081	MG461954	
MP16	2013∕7∕17	F	1,147	710	Adult	–	–	
MP17	2014∕5∕5	F	916	341	Sub-adult	HQ010081	MG461954	
MP18	2014∕5∕22	F	210	352	Sub-adult	MG461956	MG461954	
MP19	2014∕5∕30	F	752	660	Adult	HQ010081	MG461954	
MP20	2014∕5∕31	M	532	478	Sub-adult	HQ010081	MG461954	
MP21	2014∕5∕31	M	277	377	Sub-adult	–	–	
MP22	2014∕5∕31	F	734	517	Adult	HQ010081	MG461954	
MP23	2014∕6∕1	M	355	370	Sub-adult	HQ010081	MG461954	
MP24	2014∕6∕4	M	490	390	Sub-adult	HQ010081	–	
MP25	2014∕6∕8	M	296	370	Sub-adult	HQ010081	MG461954	
MP26	2014∕8∕3	F	330	366	Sub-adult	MG461955	MG461954	
MP27	2015∕5∕15	F	307	345	Sub-adult	HQ010081	MG461954	
sio07-53	2006∕11∕16	F	–	215	Juvenile	HQ010081	MG461954	
sio11-299	–	–	–	–	–	HQ010081	MG461954	

Genetic information

The cox1 gene (623 bp) and Loc6 microsatellite sequence (592 bp) were amplified and analyzed for 29 individuals obtained from Taiwan and Mexico. Three individuals failed to amplify on both loci, including MP3, MP16, and MP21, due to low DNA quality. There were two parsimony informative sites, and the nucleotide diversity (p) and haplotype diversity (h) of the cox1 gene was 0.000616 ± 0.000695 (mean ± SD) and 0.3305 ± 0.1083, respectively. Twenty-seven cox1 sequences were composed of three unique haplotypes, and the sequences from Taiwan, Mexico, Indonesia, and Puerto Rico shared a dominant haplotype (Fig. 2A haplotype network). The phylogenetic analyses showed that the sequences we used in the present study formed a monophyletic clade and that there were two nodes with substantial support, including one composed of MP2, MP7, and MP26, and the other composed of MP11 and MP18 (Fig. 2A). On the other hand, MP7 and MP24 failed to amplify for Loc6 from a sequence downloaded from GenBank derived from a Japanese specimen; therefore, a total of 25 sequences were obtained for further genetic analyses. Our results showed that the 23 sequences from Taiwan and 2 from the Mexico were identical. The haplotype derived from the Japanese coast specimen had one singleton and formed a unique haplotype separate from the dominant one. No parsimony informative sites were found, and in addition, phylogenetic analyses showed that those sequences were clustered as a single clade in the topology of the cox1 gene tree.

Figure 2 Maximum-likelihood phylogenetic trees and the median joining haplotype network based on the cox 1 gene (A) and Loc6 (B) sequence data.

Nodes are presented only for those with bootstrap scores >85% majority rule for maximum likelihood and >95% majority probabilities for Bayesian probability values (BI/ML). Different colors indicate different sampling localities (e.g., light blue, Taiwan; green, Indonesia; purple, Mexico; yellow, Puerto Rico and deep blue, Japan).

Discussion

Kuroshio as the passage to feeding grounds

More than 74% (74/99) of sighting records were from countries along the Kuroshio Current, including the Philippines, Taiwan, and Japan. Therefore, this region is likely a hotspot for the occurrence of the megamouth shark. Along the east coast of Taiwan particularly, different sizes of megamouth sharks were caught mainly from April to August off the Hualien coast (Table 1). The stomach contents of a megamouth shark caught off Ibaraki Prefecture (Japan) suggested that it fed almost exclusively on Euphausia pacifica (Sawamoto & Matsumoto, 2012). Euphausia pacifica is the dominant species of euphausiid in the North Pacific (Boden, Johnson & Brinton, 1955; Brinton, 1975) and dominates the zooplankton community in the East Sea (Sea of Japan) (Mauchline, 1980) and Yellow Sea (Yoon et al., 2000). Endo (1981) reported that the eggs and larvae of this species occur throughout the year in Sanriku waters, but are most abundant in April–June. In the Yellow Sea, E. pacifica was the most dominant euphausiid species in both summer and winter (Yoon et al., 2000). Therefore, we propose that the Kuroshio Current may be the lower latitude passage for the megamouth shark to reach its feeding grounds in higher latitudes such as the Yellow Sea and Sanriku waters where E. pacifica is abundant. Seasonal movements between productive high-latitude feeding grounds and low-latitude breeding grounds have been commonly used to explain the migration of baleen whales (e.g., Norris, 1967), and we suggest this may also explain the seasonal migration of the megamouth shark. However, a future satellite tagging study is needed to track the movement and habitat use of the megamouth shark to verify this hypothesis.

Genetic diversity and connectivity in the megamouth shark

Although the megamouth shark appears to be very rarely encountered throughout its range, IUCN assessed its population status as Least Concern based on its wide distribution (Simpfendorfer & Compagno, 2015). This rarity may lead to intrinsic sensitivity to overexploitation since the effects of genetic drift are stronger in smaller populations, which ultimately leads to a substantial loss of genetic variation (Allendorf et al., 2008) and consequently increases the probability of the fixation of deleterious alleles and reduces the resilience of overfished species (Hare et al., 2011). Genetic diversity is also one of the important indexes to be considered in shark management and conservation polices because the long-term survival of a species is strongly dependent on the levels of genetic diversity within and between populations (Domingues, Hilsdorf & Gadig, 2017). In the present study, the increasing number of captures in the Kuroshio region (Table S1), particularly Taiwan, may indicate increasing fishing pressure on megamouth sharks. Comparing its cox1 genetic diversity with other sharks (Alopias pelagicus, Scyliorhinus canicula, Squalus blainville, and R. typus; Table 2), the megamouth shark has the lowest nucleotide diversity (0.000616), and relatively lower haplotype diversity (0.3305). Among these sharks, the pelagic thresher shark (A. pelagicus) is one of the most abundant open ocean sharks and one of the most over-exploited shark species in the Pacific (Tsai, Liu & Joung, 2010; Caballero et al., 2011). Even under great fishing pressure, its nucleotide diversity was higher than that of the megamouth shark. With its rarity, increasing capture in the Kuroshio region and potentially low genetic diversity found in the present study, establishing species-specific regulations or management schemes for the megamouth shark is urgently needed.

Table 2 Examples of shark genetic diversity based on mitochondrial cox1 gene.

Species	MtDNA nucleotide diversity	MtDNA haplotype diversity	Reference	
Alopias pelagicus (Pelagic thresher shark)	0.0013	0.3066	Cardeñosa, Hyde & Caballero (2014) (Taiwan)	
Scyliorhinus canicula (Small-spotted catshark)	0.0032	0.808	Kousteni et al. (2015)	
Squalus blainville (Longnose spurdog)	0.0029	0.763	Kousteni et al. (2016)	
Rhincodon typus (Whale shark)	0.00244	0.1871	Toha et al. (2016)	
Megachasma pelagios (Mega-mouth shark)	0.000616	0.3305	Present study	

Table 3 Examples of shark population genetic structure studied at the global scale.

Species	Structure within ocean	Structure between Pacifc and Atlantic	Genetic marker	Habitat	Reference	
Carcharhinus obscurus (dusky shark)	North and South Atlantic	Yes	Control region	Reef-associated	Benavides et al. (2011)	
Carcharhinus limbatus (blacktip shark)	East and West Atlantic	Yes	Control region	Reef-associated	Keeney & Heist (2006)	
Carcharhinus plumbeus (sandbar shark)	Pacific	Yes	Control region; Microsatellite	Benthopelagic	Portnoy et al. (2010)	
Carcharias taurus (grey nurse shark)	Atlantic, Pacific, Indian	Yes	Control region; Microsatellite	Reef-associated	Ahonen, Harcourt & Stow (2009)	
Galeorhinus galeus (school shark)	North and South East Pacific, Atlantic	Yes	Control region	Benthopelagic	Chabot & Allen (2009)	
Galeocerdo cuvier (tiger shark)	North and South Atlantic, Hawaii	Yes	Control region; Microsatellite	Benthopelagic	Bernard et al. (2016)	
Sphyrna lewini (scalloped hammerhead shark)	Pacific and Atlantic	Yes	Microsatellite	Pelagic-oceanic	Daly-Engel et al. (2012)	
Rhincodon typus (whale shark)	No	Yes	Control region; Microsatellite	Pelagic-oceanic	Vignaud et al. (2014)	
Cetorhinus maximus (basking shark )	No	No	Control region	Pelagic-oceanic	Hoelzel et al. (2006)	
Prionace glauca (blue shark )	No	No	Control region; Microsatellite	Pelagic-oceanic	Veríssimo et al. (2017)	
Megachasma pelagios (mega-mouth shark)	No	No	Cox 1; Microsatellite sequence	Pelagic-oceanic	Present study	

On the other hand, information regarding population connectivity is an important consideration when establishing conservation strategies to manage threatened species. In sharks, habitat usage could be one of the major factors influencing the connectivity pattern. For example, pelagic sharks (e.g., the basking shark Cetorhinus maximus, whale shark R. typus, and blue shark Prionace glauca) that undergo long oceanic movements showed less genetic structure either within-ocean or between-ocean scales compared to coastal sharks, except that the whale shark showed a genetic break between the Pacific and Atlantic Oceans (Table 3). In the present study, neither the mitochondrial cox1 gene nor Loc6 sequence revealed any genetic structure. While a cox1 gene sequence from a specimen caught in the Caribbean was included in the analysis, it was identical to the dominant cox1 gene haplotype found in the Pacific. This suggests the megamouth shark might travel across the world’s oceans, which corresponds to its pelagic-oceanic life. Therefore, the careful tracking of fisheries captures and the implementation of a long-term global monitoring program are needed to reassess its population status and ensure that this species does not become threatened in the near future.

Conclusions

In conclusion, the Kuroshio Current region may act as a passageway for the megamouth shark to reach its feeding grounds during April to August. No genetic structure and low genetic diversity were found in the megamouth shark, suggesting a small population and the ability to travel across oceans. However, due to the small sample size and lower variability of the loci used in the present study, connectivity between sites could be overestimated. Nonetheless, to better understand the movement and migration of the megamouth shark, we recommend that in future studies the sample size be increased, hyper variable loci (microsatellite loci or SNPs) be used, and the pop-up satellite tag method be applied.

Supplemental Information

Supplemental Information 1 Aligned sequences data of COI

Click here for additional data file.

Supplemental Information 2 Aligned sequences data of Loc6

Click here for additional data file.

Table S1 Megamouth sharks recorded to date

Click here for additional data file.

We deeply thank the staffs of George Chen Shark Research Center, National Taiwan Ocean University who helped to carry out the biological measurements in the field.

Additional Information and Declarations

Competing Interests

Author Contributions

Animal Ethics

DNA Deposition

Data Availability

The authors declare there are no competing interests.

Shang Yin Vanson Liu conceived and designed the experiments, performed the experiments, analyzed the data, contributed reagents/materials/analysis tools, prepared figures and/or tables, authored or reviewed drafts of the paper, approved the final draft.

Shoou Jeng Joung conceived and designed the experiments, contributed reagents/materials/analysis tools, authored or reviewed drafts of the paper, approved the final draft.

Chi-Ju Yu, Hua-Hsun Hsu, Wen-Pei Tsai and Kwang Ming Liu performed the experiments, contributed reagents/materials/analysis tools, authored or reviewed drafts of the paper, approved the final draft.

The following information was supplied relating to ethical approvals (i.e., approving body and any reference numbers):

We sampled vertebrates from fish markets as fishery landings and therefore no IRB approval was needed for the study.

The following information was supplied regarding the deposition of DNA sequences:

GenBank Accession numbers MG461954, MG461955 and MG461956.

The following information was supplied regarding data availability:

The sequence datasets used in this study are provided as Supplemental files.

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
