# Peer review of "Genetic diversity and connectivity of the megamouth shark (Megachasma pelagios)"

_PeerJ, doi:10.7717/peerj.4432_

## Round 0.1 · original submission · Major Revisions

I have heard back from two reviewers, both of whom offer extensive comments on your work. Reviewer 1 stresses the need for reconsideration of some of your analytical methods, while reviewer 2 mentions the benefits of using peer-reviewed data in your analyses. As well, both reviewers have made comments on figures, tables, labelling, and some English brush-ups. Thus, overall, while interesting, it is my judgement that your paper needs a fair amount of work and reconsideration before being acceptable; please take the time to carefully consider all comments while revising your paper.

Reviewer 1 ·

Basic reporting

The paper presents the first data on the genetic structuring patterns of the megamouth shark (Megachasma pelagios) using sequence data generated from the mtDNA COI gene and a nuclear microsatellite locus (Loc 6). Indeed, the megamouth shark is rare and obtaining adequate samples for a population genetics study is challenging; however, inadequate sampling compromises the population genetics analyses and interpretation of results.

Overall, the paper reads well although it could do with re-structuring of several sections such as the Methods and Discussion. Kudos to the authors for a thorough literature review on shark population genetics, although it is disappointing to note that the author did not use the available information when selecting their molecular markers.

The figures provided by the authors are of low quality and need minor edits. The authors are encouraged to check their tables again.

I suggest that the paper is re-focused on the patterns of genetic diversity distribution, without making firm statements on the effects of biogeographic barriers on the genetic structure of the megamouth shark. This latter issue requires larger sample sizes for each studied sampling population.

Experimental design

I have many doubts on the adequacy of the sampling effort to achieve the set objectives stated in the last paragraph of the Introduction; “…to test (1) the effect of the East Pacific Barrier and closure of the Isthmus of Panama on the genetic structure of the megamouth shark and (2) whether it can travel from ocean to ocean.” With an overall sample size of 27 individuals the genetic diversity of the putative populations cannot be adequately represented.

Apart from issues surrounding sampling sizes per location, my criticism is on the choice and the polymorphism levels of the markers of choice, which are concerning. When sampling sizes are low, often, one screens several DNA markers to identify a highly polymorphic one. It is confounding that the authors would use a DNA Barcoding region as a marker of choice. Why? Moreover, why sequence a microsatellite locus?

Interestingly, the author constructed a table (Table 3) detailing examples of shark population genetics studies at a global scale, where the principal marker of choice was the mitochondrial control region. Within Table 3 the authors note that the control region and microsatellites were used in the present study; however, this is not true. Because of the low sample size and the lack of polymorphism of the both the COI gene (as to be expected intra-specific) and the nuclear microsatellite loci, the resolution is very low, which resulted in the failure to detect genetic differentiation.

Validity of the findings

Given the low resolution of the markers used in the present study several analyses that were done are flawed. First, the pairwise FST index cannot be conducted with such a low sampling population sizes. Second, the use of Bayesian Skyline Plot (BSP) analysis is only appropriate for informative enough data sets, thus the results obtained for such an analysis are misleading and SHOULD be omitted. Last, although no described in the Methods section, the use Tajima’s D test is also not appropriate, and the results are misleading. Frankly, I believe that all the interpretation based on all these analyses are in many ways ad hoc and speculative considering the intrinsic properties of the set of markers from which the results are derived. Also, it is not clear as to why the authors constructed phylograms using both Bayesian and Likelihood approaches. The logic escapes here.

Additional comments

Line 96: This does not make sense, unless a same individual was sampled more than once, and the duplicates removed. Please correct this.

Lines 155-157: Likewise, the mathematics does not check out here based on Line 96. Please amend accordingly.

Line 201-236: I fear that this entire section is detrimental to objectives of the study. All the discussion trying to link the lack of genetic structure with the ineffectiveness of the closure of the Isthmus of Panama in restricting gene flow is, in my opinion a bit confusing and speculative.

Lines 237-251: This section is not necessary, and it is also detrimental to objectives of the study, and I encourage the authors to remove it.

Reviewer 2 ·

Basic reporting

1. Make sure to double check English language use, although most of it is fine some of the following corrections are slight improvements on wording to strengthen points and English language use, while others might help clarify

Line 22: You have mentioned no other proxies for indirectly measuring gene flow, consider removing another “This approach provides a proxy to indirectly measure gene flow between populations.”

Line 39: Consider rewording “… only about 120 specimens have been caught or documented.”

Line 40-41: “The only official published record is for Japan (Nakaya, 2010), which documents 40 records …”

Line 43: “The latest record is No. 117, from Komodo, Indonesia, in July 2017.”

Line 44: sources may be a more appropriate word than documents

Line 49: replace during with from

Line 49-52: consider breaking this up into multiple sentences to make the point more clear, it becomes a little confusing.

Line 56-58 Consider rewording “In addition, due to the scarcity and vulnerability of these sharks, satellite tagging has not yet been feasible. Therefore, information about their horizontal…”

Line 64: has frequently been found or is frequently found

Line 68-70—Sounds a little awkward, consider rewording. Also, maybe specify that Taiwan instated a National Plan of Action concerning sharks in 2006

Line 93: GenBank

Line 111: remove second gene

Line 123-124: “Two cox1 gene sequences of individuals from Indonesia (EU3938905) and Puerto Rico (KY392958.1) were downloaded from GenBank. In addition, a Loc6 sequence derived from a Japanese specimen was downloaded.” Or something similar, currently it

Line 130 and 131: in text citation format changes

Line 164: replace notes with nodes

Lines 181-184: Awkward phrasing... consider revising

Lines 202-203: Consider restructuring this sentence

Line 222: include Rhincodon typus

Line 223: remove in “…showed no genetic structure within or between oceans, except the whale shark which showed…”

Line 226-231: Maybe divide into two sentences. “…gene haplotype found in the Pacific. This indicates that the…”

2. Figures and Tables are nice, but could use some improvement to adhere to standards. At this time the quality of some is lacking, and descriptions can be improved.

Figure 1: This figure could use some improvement, consistency with labeling would help, either do specific locations or country where they were caught, the combination could become confusing (for example have all of the labels as Japan, Indonesia, Mexico, Taiwan, Puerto Rico). Additionally, due to the focus on the Kuroshio region, it may be helpful to highlight this area on the map. The map in general could use some improvements, for the projection being used, the scale would not be consistent throughout the map.

Figure 2: Figure caption needs to be more descriptive, it provides no information about the figure, although there is description within the text this should be stand alone. Please specify that which tree and network A and B are, etc. In addition, both the PDF version and eps file are blurry and difficult to read. Specify which order your values on tree A are in ML/BI or BI/ML


3. Make sure to double check the references. Line 364 is not correct, and I think Line 376 might be part of that reference.

Experimental design

1. Research Question
I think the first objective is very reasonable and that the paper covers results from this objective/question well.

However, Line 91: objective 2--- I’m not sure this is the best way to phrase the question, because you aren’t really examining if they can move from one ocean to another, you are looking more at genetic connections and structure between the ocean basins. Although this may suggest that they can move from ocean to ocean, without the tracking data this isn’t really the point you working towards. Consider rephrasing this.

2. Methods: It might be useful to define what differentiates the juveniles, sub-adults, and adults. What qualifiers were used in order to make this determination? This is especially relevant since this is brought up in the results.

Validity of the findings

For the majority, the findings are good and provide insight into popgen of a poorly understood species.

1. For the assessment of catch information: It is stated that megamouth sharks were generally caught between April and July, however there is no mention of when the fishing effort occurs. Is there fishing year round or is there only effort during that season? It may be out of the immediate scope of this paper, but knowing even rough catch per unit effort or if there is fishing in the areas where they are being caught year round may be helpful and would help to support your seasonality hypothesis. I completely understand that you may not have this data, but it may be worth considering to help strengthen your point about seasonality.

2. The conclusions are well stated. I think that there is an air of caution and the limited sample size's potential to bias the results and limitations of the markers are expressed. However, this is once again where I feel that the second objective listed in the introduction is left out, and would warrant it either being removed entirely or stated in a different manner and discussed more.

Additional comments

I commend the authors for making the most of a limited data set and limited samples, I think it is a useful look into megamouth sharks that has the potential to be built on further throughout time.

However, I am concerned about the reliance on “sharkmans-world.eu” website for some of the information concerning occurrences of these sharks. I realize it may be the only source that comprehensively documents the occurrence of the sharks, but I would be more comfortable with more references to peer reviewed documentation (for example the listing of the first 14 occurrences in Amorim, Arfelli, & Castro, 2000) . Although it is considerably more work, it may be better to use previous peer reviewed reports and the direct sources of the sightings and create a supplementary table that creates a comprehensive source within this publication. In my opinion it would be a more compelling and trusted source than the webpage and would most likely be highly used in the future as well.

---

## Round 0.2 · Minor Revisions

Your submission has been greatly improved, but there are some final minor comments that need attention. I would like to stress the importance of making sure all the English is correct as we do not provide editing services at PeerJ; the onus is upon you and your co-authors.

I look forward to seeing a revised version of your work.

Reviewer 1 ·

Basic reporting

The paper is much improved and focuses on reachable objectives given the small sample sizes.

Experimental design

No comment.

Validity of the findings

No comment.

Additional comments

I congratulate the authors for their erudite contributions to the field of shark genetics and, providing the first and valuable genetic knowledge base on the megamouth shark.

Reviewer 2 ·

Basic reporting

The paper reads quite well and improvements have been made. However, I still think some of the English could stand to be improved, especially within some of the areas where additions and changes have been made.

Line 39: Consider getting rid of “tracking”
Line 45-47: Consider rephrasing to improve the flow.
Line 82-83: Sounds awkward, consider rewording
Line 85-86: “…collected from Baja California, Mexico; 27 tissues samples from Hualien, eastern Taiwan; and published…”
Line 198: consider rephrasing to either “… genetic diversity to that of other sharks…” or “…genetic diversity with other sharks…”
Line 212: “which undergo” instead of undergoing
Lines 227-229 are awkward, consider rewording.



Line 188-189: citation for Red List assessment is missing (also not listed in the reference list). Should be Simpfendorfer, C. & Compagno, L.J.V. 2015


Supplementary Table: A great improvement, and useful resource. However, it feels out of place when mentioned in Lines 145-147 (as part of the results). Perhaps add reference to the supplementary table early on in the introduction (around line 47) when the resources sighted there are heavily referred to.

Figure 1: This has also improved; however the scale bar should be removed, it is not accurate for the map projection being used, and in this case is not necessary.

Experimental design

This area has had the most improvement. I think the new direction was a great suggestion and was beneficial to the manuscript.

The new more reserved analysis of diversity and connectivity is more appropriate.

Validity of the findings

This has also improved, although I think it is important to continue to use great caution and not try to oversell the results too much, considering markers used and small sample size. In the conclusions these limitations are clearly pointed out. I am concerned about Line 217-218, I think it would be more beneficial to cautiously suggest the possibility of connections between the oceans. The current sentiment may be a misleading or slightly overstated.

Line 201-207: I find this section a little worrisome. Although the table of comparisons to other COI studies is interesting, the direct comparison to the A. pelagicus study seems to be a stretch. I am especially considering the drastic difference in sample size and number of sampling locations between the two studies. This caveat should be clearly explained or comparisons should be cautiously made. Also, the final sentence is not entirely clear and could use a bit of work.

Additional comments

Kudos to the authors for the great improvements that have been made.

Please make sure to double check formatting within the tables and figures, there still seem to be some minor details that need to be attended to. In Table 3 one of the scientific names is not italicized (whale shark). In Table 1, Date of Collection would sound better than Date of Collecting.

The use of the Florida Natural History Museum is a vast improvement over the previous website.

---

## Round 0.3 · accepted · Accept

Your work has been well revised - good work! Please note there are some final edits I have made on the revised English (see MSWord file) - please ensure all revisions are done no later than the proof stage.